# Seismic Monitoring of a Deep Geothermal Field in Munich (Germany) Using Borehole Distributed Acoustic Sensing

**DOI:** 10.3390/s24103061

**Published:** 2024-05-11

**Authors:** Jérôme Azzola, Emmanuel Gaucher

**Affiliations:** Karlsruhe Institute of Technology (KIT), Institute of Applied Geosciences (AGW), Adenauerring 20b, 76131 Karlsruhe, Germany; emmanuel.gaucher@kit.edu

**Keywords:** DFOS, distributed acoustic sensing, deep geothermal energy, seismic monitoring, geothermal reservoir, Molasse basin

## Abstract

Geothermal energy exploitation in urban areas necessitates robust real-time seismic monitoring for risk mitigation. While surface-based seismic networks are valuable, they are sensitive to anthropogenic noise. This study investigates the capabilities of borehole Distributed Acoustic Sensing (DAS) for local seismic monitoring of a geothermal field located in Munich, Germany. We leverage the operator’s cloud infrastructure for DAS data management and processing. We introduce a comprehensive workflow for the automated processing of DAS data, including seismic event detection, onset time picking, and event characterization. The latter includes the determination of the event hypocenter, origin time, seismic moment, and stress drop. Waveform-based parameters are obtained after the automatic conversion of the DAS strain-rate to acceleration. We present the results of a 6-month monitoring period that demonstrates the capabilities of the proposed monitoring set-up, from the management of DAS data volumes to the establishment of an event catalog. The comparison of the results with seismometer data shows that the phase and amplitude of DAS data can be reliably used for seismic processing. This emphasizes the potential of improving seismic monitoring capabilities with hybrid networks, combining surface and downhole seismometers with borehole DAS. The inherent high-density array configuration of borehole DAS proves particularly advantageous in urban and operational environments. This study stresses that realistic prior knowledge of the seismic velocity model remains essential to prevent a large number of DAS sensing points from biasing results and interpretation. This study suggests the potential for a gradual extension of the network as geothermal exploitation progresses and new wells are equipped, owing to the scalability of the described monitoring system.

## 1. Introduction

Geothermal reservoirs are dynamic underground fluid circulation systems subject to variations in temperature, pressure, and stress. The exploitation of deep reservoirs can lead to induced seismicity even in hydrothermal reservoirs, which are mainly driven by a porous rock matrix. Such conclusions have been drawn for instance in the Munich area, which is the central focus of this study [1,2,3]. Geothermal energy has been harnessed in this region since the late 1990s [4,5] and multiple projects are underway to further exploit this renewable heat source [6]. The growth of this energy sector underlines the importance of sustainable development and operation of all current and prospective geothermal fields, especially in urban areas. To achieve this goal, geothermal plant operators wish to implement efficient, reliable, and integrated reservoir management systems [7]. Tailored real-time seismic monitoring for seismic risk mitigation would constitute an essential component of such reservoir management systems. 

Seismic monitoring based on individual—mainly surface—discrete three-component sensors provides valuable insights but faces sensitivity limitations in urban environments due to anthropogenic noise. Hence, it may be appropriate to take advantage of the drilling stage at any site undergoing development. Each newly drilled well could become an integral part of the monitoring network, facilitating its expansion in proportion. In this view, the deployment of fiber optic cables (FOCs) constitutes a robust and promising solution for borehole sensing. In recent years, deep geothermal operators have invested in Distributed Optical Fiber Sensing (DOFS) to assess the benefits of the technology. It leverages optical fibers to deliver repeatable in-situ observation of changes in temperature, strain and/or strain-rate. DOFS applications demonstrated that it could provide insight into, e.g., in-flow and out-flow zones, fluid flow rates, well integrity, and induced seismicity [7,8,9,10].

Distributed Acoustic Sensing (DAS) is applied along the optical fiber to monitor induced seismicity. It consists of measuring the strain-rate (SR) along regularly distributed portions of the fiber and over a large frequency range, including the frequency of seismic signals [11]. More details on the principle of this sensing technique can be found in e.g., [12,13]. In addition, a broad review of applications can be found in [14]. The use of a FOC that is securely cemented behind the casing is particularly advantageous. It prevents the recording of noise generated by a flowing borehole while ensuring unimpeded well operation and allows continuous and repeatable data acquisition [15]. It also creates a high degree of coupling between the sensor and the subsurface [16]. Borehole DAS provides a substantial set of sensing points (SP) closer to the geo-reservoir, which is the primary target of the monitoring. Additionally, it reduces the impact of surface noise on these data [15,17,18], which increases the sensitivity of monitoring systems in noisy operational or urban environments. Finally, a broad range of borehole seismic processing techniques may be accessible for DAS data de-noising [19] and event detection [20,21]. 

In this study, we use DAS at the deep geothermal site of Schäftlarnstraße (SLS), Munich (Germany), for local seismic monitoring to supplement the capabilities of the public seismic network. At this geothermal site (Figure 1), a FOC was installed in the injection well TH3 [22]. Previous work demonstrate that DAS monitoring at this site is able to detect seismic events not identifiable by standard surface or shallow-borehole 3C-seismometers [17], thereby enhancing the seismic detection capabilities. [17] addresses the feasibility of setting up an infrastructure capable of managing the large data flow generated by the DAS, continuously and in near-real time, for the detection of local seismic events. This infrastructure is built on a cloud Internet of Things platform, and it uses the operator’s IT resources for optimized data integration. It features a secured storage environment for these DAS data and scalable processing resources (Figure 2).

The primary focus of [17] is on the development and integration of the infrastructure for quasi real-time seismic detection, providing a relatively limited seismic interpretation of detected events. However, accurately resolving the location, magnitude, and stress drop of seismic sources is key to seismic monitoring. The capacity of the DAS monitoring system to provide quantitative analysis of strain-rate data was yet to be assessed, and different challenges can be outlined to achieve this objective. Seismic source models are based on ground motions, i.e., displacements, velocities or acceleration [23,24] and DAS records strain-rate. The application of established methods based on these models requires, therefore, prior conversion of the DAS strain-rate to ground motion. For further waveform processing, correct phase and amplitude recovery must also be guaranteed. In recent work, Ref. [25] or [11] demonstrated that assuming no or minimal phase distortion from the DAS interrogators is a valid approximation of the instrument’s phase response. The system amplitude response, despite being flat in a large frequency range, is more sensitive to experimental conditions. The strain recorded along the fiber may not represent the actual rock strain because of factors such as the coupling of the fiber to the ground or the internal structure of the FOC. Lastly, DAS provides an unprecedented density of sensing points. When they are combined with sparser observation points, such as the seismometer network, their relative weight in the calculations requires particular attention, especially with regard to the possible bias in the results.

This study addresses the aforementioned challenges. It introduces additional features to the monitoring system proposed by [17] to provide, in a timely manner, a comprehensive catalog of local seismic events. Hence, this study presents the capabilities of monitoring and characterizing seismic activity using borehole DAS technology at an active geothermal field. Section 2 of the manuscript presents the context of this study, including the geothermal site and the cloud infrastructure, which is developed to store and process these DAS data. In Section 3, we detail the workflow to process the DAS data automatically. It covers multiple aspects of seismic monitoring, including seismic event detection, accurate picking of the seismic wave onset times, location, determination of the scalar moment (*M*_0_), moment magnitude (*M*_W_), and stress drop (∆S). In Section 4, we illustrate the results with two local events, which were detected during a 6-month continuous monitoring period that aimed at assessing the DAS-based monitoring concept. These results are further discussed in Section 5. We particularly discuss the influence of DAS observations in locating seismic events, as the technology provides a high density of unidirectional sensing points along the TH3 vertical FOC. We also evaluate the phase and amplitude of converted DAS waveforms in view of data recorded by a nearby seismometer station.

## 2. Study Site and Monitoring Infrastructure

In this section, we provide the fundamental background information for this study. For a comprehensive understanding of the research framework and background, readers are encouraged to refer to [17] who also provide an extensive description of the infrastructure initiated for DAS monitoring at the geothermal field.

### 2.1. Geothermal Field and Setup for Fiber Optic Sensing

The geothermal field developed at Schäftlarnstraße (SLS) by Stadtwerke München GmbH (SWM) aims at meeting the heating needs of around 80,000 inhabitants and plays a crucial role in Munich city’s transition to renewable energies. Six wells give access to the geothermal resource, i.e., three pairs of injection and production wells. The trajectories, which are projected on the city’s map in Figure 1, are vertical at shallow depths and start to deviate from about 800 m below ground level (bgl). The SLS wells target the Malm carbonates, which present favorable hydrothermal characteristics for heat extraction [26,27]. The Malm formation is the primary geothermal reservoir in the Molasse basin and it is crossed by all wells between approximately 2500 and 3000 m [27,28]. 

Monitoring local induced seismicity in the area is critical for safe and sustainable development of geothermal energy projects [2]. The growing presence of deep geothermal fields in the Greater Munich urban area necessitates the establishment of specialized, local seismic monitoring systems to supplement public monitoring infrastructures. Figure 1 shows the network of 11 broadband seismometers used in this study and recording in a 15 km radius around SLS. The red dots show the location of five stations installed in the frame of the INSIDE project (BMWK project 03EE4008C), and the green dots show the location of stations part of the BayernNetz [29]. The location of the study site, in the city of Munich, implies a significant level of seismic noise, mainly attributed to anthropogenic activities. Hence, the seismometers closest to the geothermal field, namely SYBAD and SIEM (Figure 1), are 3C-borehole seismometers located approximately 180 m deep in shallow water-table monitoring wells. All other sensors included in Figure 1 are installed at the surface. The sensors installed within the INSIDE project record at a sampling rate of 250 Hz. The sampling rate of the other sensors is 200 Hz.

Fiber optic cables have been deployed at SLS to assess the benefits of DOFS in the context of geothermal energy applications. We focus here on measurements collected from the FOC cemented behind the casing in the first section of the TH3 injection well (Figure 2). Injection in the Malm reservoir takes place between 2570 and 3049 m. The trajectory of the optical fibers in the well is assumed to be vertical and straight, from the surface down to a micro-bend located at 692 m. In the present study, a Febus Optics A1-R interrogator was connected to a tight mono-mode fiber for continuous DAS acquisitions. The micro-bend created at the bottom end of the FOC allows the connection of optical fibers in a U-loop configuration. It enables the transmission of the laser pulse downwards and then upwards for repeated sensing capabilities from a single cable end. A gauge length of 10 m, a sampling distance of 2.5 m, and a sampling rate of 500 Hz were applied. As a result, analyzed DAS datasets consist of 280 strain-rate time series (the traces), each corresponding to sensing points (SP) distributed along the initial 692 m of TH3. [22] provide a comprehensive description of the on-site FOC equipment and [17] give additional details on how the position of each SP along the FOC was defined from tap-tests.

### 2.2. Infrastructure for Data Saving and Processing

Continuous and permanent DAS monitoring requires an adapted infrastructure for transferring, saving, and processing the generated data. To ensure effective seismic monitoring, the infrastructure is expected to fulfill the following requirements: It should guarantee reliable and secure connectivity for data transmission and protection.It should have sufficient storage capacity and efficient data management systems to handle the large volume of data generated.It should feature computational capabilities for data processing and analyses, providing quasi-real-time processing capabilities.Additionally, the infrastructure should be scalable to accommodate forthcoming technological advancements, methodological innovations, and the evolving requirements set forth by the operator.

Accordingly, a cloud-based infrastructure was implemented and tested at SLS between February and August 2022 (Figure 2). It harnesses tools and services provided by the geothermal field operator, SWM, and gives the possibility to merge different monitoring components into a cohesive cloud infrastructure. In particular, it lays the groundwork for combining seismic and plant operation monitoring, offering valuable support for establishing a sustainable reservoir management system [7,17]. The cloud platform gives access to scalable storage resources in the form of a so-called Data Lake. The cloud-based workstations provide scalable computational resources to process incoming data, with the primary objective of detecting local seismic events with minimum delay. Hence, this cloud-based approach allows efficient data management and triggering while ensuring secure and remote access to these DAS data. The suitability of the infrastructure for near real-time detection of seismic events was discussed in [17]. In this study, we extend the data processing functionalities of the monitoring system using both local and cloud-computing capabilities (Figure 3). Following event detection, onset times are measured trace-by-trace for P- and S-waves using an automated picking algorithm running on cloud workstations (see Section 3). The development of the processing capabilities also involves the use of local workstations for event location by third-party software [30,31]. The resource optimization to minimize the delays in obtaining seismic source characteristics is addressed in Section 5.

## 3. Seismic Processing Workflow

Figure 3 summarizes the workflow applied to any DAS data files landing on the Data Lake. The first step consists of pre-processing DAS data on cloud workstations. The second step aims to use the structured DAS datasets for the detection of potential seismic events within a 15 km radius. The final step is dedicated to the characterization of seismic sources. This section details the seismic processing functionalities highlighted in red in Figure 3. For a more comprehensive understanding of the pre-processing and detection functionalities, readers are referred to [17].

### 3.1. Event Detection and Automated P- and S-Wave Arrival-Time Picking

The pre-processing of the data files covers the loading and structuring of these data (see Figure 3). Data structuring includes the extraction of the 280 SP distributed along the first 692 m of TH3, the assignment of the physical position and the definition of an Obspy stream [32] containing the 280 traces. The Obspy stream, with a spatial sampling of 2.5 m and a temporal sampling of 2.0 ms, then enters the detection workflow. The traces are first filtered using a forward and backward fourth-order Butterworth band pass (BP) filter [33]. It is applied between 5 and 40 Hz, a typical frequency range for the detection of local induced seismicity (e.g., [34]). The dense spatial sampling and spatial coherence of DAS data enable additional filtering in the frequency-wavenumber (f-k) domain. As discussed by [35], f-k filtering allows removing energy related to identified disturbances and incoherent noise, or isolate spatial features of interest in seismic data. The filtering first converts the signal from the time-offset domain to the frequency-wavenumber domain. Then, it sets the energy associated with the perturbation(s) to zero to remove and, finally, reverts the signal to its original time-offset domain. For event detection, we design an f-k filter to preserve only up-coming waves, i.e., waves propagating from the deepest part of the fiber to the surface (top-right and bottom-left quarters in the f-k representations). This implies that we focus on events that occur below the fiber’s micro-bend, i.e., below 692 m. This f-k filter decreases noise originating from surface operations and eliminates laser noise [36]. It results in an overall improvement in the signal-to-noise ratio, as illustrated in Section 4. After filtering, the detection of events is based on a network coincidence approach implemented in the Obspy library [32] and using a recursive STA/LTA algorithm [37,38]. This procedure is independent of the velocity model. 

Several regional and two local seismic events have been detected during the testing period planned between February and August 2022. To assess the capability of the DAS monitoring infrastructure, we focus here on the two local seismic events presented in Figure 4. The figure includes the associated BP and f-k filtered strain-rate traces, where a clear signature of the first P- and S-waves is visible on the DAS recordings. These two seismic events illustrate noteworthy scenarios for local seismic monitoring at SLS. One event was clearly identified by the local seismometer network (Figure 4a). It occurred on 9 February 2022, and its local magnitude was estimated to be 1.5 from the seismometer network. The second event was only visible at the SYBAD station, i.e., from the downhole 3C-seismometer located about 1 km east of the TH3 wellhead (Figure 1). It took place on 22 April 2022 (Figure 4b), and the characteristics of the seismic source could not be determined from the seismometer network due to insufficient data collected. Figure 4 also shows unfiltered traces in the f-k domain for both events (middle panels). The signature of the laser noise is visible in both cases, with consistently high amplitudes along the horizontal axis, corresponding to an infinite apparent velocity. Surface activity, producing downwards propagating vibrations, is observed in the positive frequency/negative wavenumber or negative frequency/positive wavenumber quadrants (lower-right and top-left quadrants). The P- and S- waves propagating from the deepest part of the FOC to the surface are clearly identified in the positive frequency/positive wavenumber or negative frequency/negative wavenumber quadrants (top-right and bottom-left quadrants). 

Figure 4 also shows the results of the automated onset-time picking (shown as red lines on the left). This process involves a narrow f-k filtering aimed at isolating up-going P- and S-wavetrains with apparent velocities ranging from 1600 to 3500 m/s and 500 to 1600 m/s, respectively. These limits are highlighted in the f-k domain. Filtered wave fields are additionally shown in Appendix A for the 9 February (Figure A1) and the 22 April (Figure A2) events with the corresponding correlation diagrams. The latter diagrams illustrate the spatial coherence within the datasets by quantifying the correlation observed between each pair of DAS traces. For both events, we observe a significant spatial coherence, especially at greater depths where the surface noise has been attenuated. This characteristic is used for the automatic onset-time picking. In this procedure, we first estimate the arrival times by applying a STA/LTA algorithm [37,38] to the dataset narrowly filtered in the f-k domain, for either P- or S-waves. Second, on the trace with the highest signal-to-noise ratio (SNR), the SR standard deviation prior to the STA/LTA pick is calculated and used to determine the earliest preceding zero-crossing time. This time is assigned to the wave onset time on the trace with the highest SNR. Finally, a segment of the waveform around this onset time is extracted and cross-correlated with all other traces of the dataset. As a result, the onset time is propagated along the entire length of the optical fiber. The uncertainty of the onset times is set to 1/4 the signal main frequency, resulting in slightly lower uncertainties compared with manual picking on the surface seismometers. This observation is consistent with the higher frequency content typically observed in borehole measurements compared with surface measurements.

The final step of the detection workflow is the archiving of the processing results on the cloud infrastructure (Figure 3). They consist of a preliminary catalog of detected events in XML format that includes all onset-times and the associated waveforms in miniSEED format [39].

### 3.2. Seismic Source Characterization

Post-processing of the detected events is carried out on local workstations using dedicated Python scripts. The workflow is automatically triggered after the addition of new detection results on the Data Lake. After the event location, the seismic source characterization aims to evaluate the scalar moment (*M*_0_), the moment magnitude (*M*_W_), and the stress drop (∆S). Estimating the latest three parameters necessitates converting the strain-rate traces into ground motion.

#### 3.2.1. Event Location

Event hypocenters are computed using the P- and S-wave onset times measured by DAS and at other seismometer stations (see Section 4.2 and Section 5.1). We use the NonLinLoc software package (V7.00) [30,31,40] and locate in the most comprehensive 3D-velocity model available for the study area. This model is based on the main lithological interfaces of the geological structure obtained through 3D-seismic migration, supplemented by acoustic logs and Vertical Seismic Profile (VSP) data obtained from geothermal wells within the area of interest. Additional details about the procedure followed to construct the velocity model can be found in Appendix A (Figure A3 and Text A1). Weighted least-square minimization of the arrival time residuals is applied in this 3D-velocity model to determine the source location [41,42]. The minimization procedure is based on the Oct-Tree grid-search algorithm [40] that recursively subdivides the 3D model in child-cells, converging towards the region of maximum likelihood using a tree structure. In addition to the most likely hypocenter, importance sampling yields probability density functions (PDF), thus location uncertainties.

#### 3.2.2. DAS Strain-Rate Data Conversion

Conventional methods for the evaluation of source parameters are based on source models relying on ground motions, i.e., displacements, velocities, or accelerations [23,24,43]. Hence, DAS strain-rate data need to be converted to the true ground motion before the application of these well-established methods. DAS data conversion can involve a collocated seismometer [44]. It is used as a reference point for spatially integrating SR data. This methodology cannot be applied here. Alternative observation-based approaches consist of estimating the apparent phase velocities along the fiber and converting SR to acceleration in the time domain using Equation (1), where *ε*, *A* and *s* represent strain-rate, acceleration, and apparent phase slowness along the fiber, respectively:*ε*(t) = *A*(t) × *s*(t)(1)

Due to the wave-paths’ complexity, apparent velocities are subject to rapid variations, and imposing a constant value over time would bias the conversion to ground motion (see Figure A4 and Figure A5 and Table A1 in Appendix A and discussion in Section 5.2). Hence, we follow the procedure introduced by [45] to estimate a full time-series of slowness values over the SR signal. The authors validated their procedure using both synthetic measurements and ocean bottom FOC observations. Results obtained by this approach on the SLS data are illustrated in Section 4.2. 

For each individual trace of the DAS dataset, a slant-stack transform is performed at a local scale to infer the evolution of the apparent phase velocity over time. For this purpose, a data subset centered on the trace to be converted is first selected. The spatial extent of this window, *L*, should exceed the gauge length and be selected to ensure sufficient spatial coherence between adjacent traces. As shown by [17], these requirements are satisfied by applying *L* = 100 m, which corresponds to a window of 41 successive traces. The waveform semblance is computed over time after shifting the traces of this subset using their distances to the trace to be converted at the center of the subset and the selected slowness value. This process is iterated over slowness values. Then, for each time step, the slowness associated with the largest semblance value is extracted and kept for data conversion. A moving average with a window size set to the signal’s longest period of interest is applied to avoid sharp variations in the slowness of the time series. The previous steps are repeated for each trace of the DAS dataset.

#### 3.2.3. Source Parameters Evaluation

Source parameters are inverted by adjusting an omega-squared model [46] to observed displacement amplitude spectra. The latter are computed for each trace using 0.8 s long data windows starting at the P-wave onset time. The synthetic acceleration spectrum $\ddot{Ω}$ is given by Equation (2), where Ω_0_ is the low-frequency displacement spectrum plateau, *f* is the frequency, *f*_0_ is the source corner frequency, and *f*_k_ is the attenuation corner frequency controlling the drop-off at high frequencies.
(2)$$\ddot{Ω}=2\pi f^{2}\;\frac{{Ω}_{0}}{1+{\left(f/f_{0}\right)}^{2}}\exp\left(-f/f_{k}\right)\;\;$$

For the analyzed body waves, the low-frequency spectrum plateau is considered proportional to the scalar seismic moment *M*_0_:(3)$$M_{0}={Ω}_{0}\;\frac{4\pi \;\rho \;V^{3}D}{U\;F_{S}}$$

In Equation (3), *V* is the body-wave propagation velocity, *D* is the station-to-source distance, *U* is the radiation pattern term, and *F_S_* is the free-surface coefficient. A free-surface correction factor *F_S_* = 2 is applied to surface stations. However, for the borehole measurements, i.e., for DAS in TH3 and for the borehole stations SYBAD and SIEM, *F_S_* = 1 because the depth of the sensors is considered larger than the P-wavelength. To overcome the unavailability of the *U* value in Equation (3), we use the mean radiation term over the focal sphere, which is 0.52 for P-waves. The moment magnitude is subsequently computed from the scalar seismic moment *M*_0_, according to [47].
(4)$$M_{W}=\frac{2}{3}\;\left(\log_{10}\left(\;M_{0}\right)-9.1\right)$$

In addition to the seismic moment *M*_0_, additional information from the source are necessary to report on the seismic event dynamic. To distinguish between small slips on large faults and large slips on small faults, we evaluate the stress drop ΔS assuming a circular fault model, as proposed by [24,48].
(5)$$∆S=\frac{7}{16}\;M_{0}\;{\left(\frac{f_{0}}{k\;V_{S}}\right)}^{3}\;\;$$

In Equation (5), *f*_0_ is the corner frequency. Assuming that the rupture velocity is 90% of the shear velocity [24], we apply *k* = 0.32 for P-waves. The equation highlights that the stress drop is proportional to the cube of the corner frequency. Consequently, the accuracy of the stress drop strongly relies on the accurate determination of the corner frequency, which, in turn, depends on the selected fault model and the spectra representativeness. To improve the latter and mitigate unfavorable SNR conditions for the DAS dataset, we estimate the source parameters from average amplitude spectra. They are computed by averaging individual spectra calculated across a gauge length, i.e., five spectra spanning 10 m. In addition, we use the spectrum of each trace only once. The source parameters are consequently computed and sampled every 12.5 m along the optical fiber in TH3.

## 4. Results

### 4.1. Picking of Arrival-Times

To assess the consistency of the P- and S-wave onset times presented in Figure 4 (left panels), the measurements are compared with manual picks obtained using the Reveal© software (V.5.2, Shearwater). Along the first 100 m for the February event and the first 150 m for the April event, the automatic picking fails in capturing accurate onset times, which is the consequence of poor inter-channel coherence. This lack of coherence is attributed to higher background noise levels and interference between upwards- and downwards-propagating waves. The results for traces deeper than 150 m are shown in Figure 4 (right panels) in the form of box plots. It shows that the automatic picks deviate consistently from the manual ones. This delay varies depending on the central frequency of the analyzed waves, with a larger delay observed for the S-waves compared with the P-waves. Automatic and manual picks deviate by 3 to 4.5 ms for the P-waves. The maximum shift is about 6 ms, which corresponds to 1.5 samples (at 250 Hz). For the S-wave, the systematic deviation ranges between 10 and 15 ms, and the maximal difference is close to 25 ms. This delay remains, however, small compared with the period of the analyzed waves, which is about 125 ms for the S-wave of the 9 February event. Moreover, automatic and manual picks yield consistent hypocenters. For the event on 22 April, switching from automatic to manual onset times results in shifting the depth of the hypocenter by 85 m. This difference falls within the range of inversion uncertainties, described in the following, which demonstrates the applicability of the automatic picking approach.

The filters designed in the f-k domain (see Section 3.1) significantly improve the SNR, thereby contributing to seismic event detection and onset-times picking. Figure 5 highlights that keeping the up-going wavefield performs better than just applying the BP filter (panels (a) and (b), blue vs. red dots). The SNR improvement is emphasized when using the narrow f-k filter that focuses on the range of apparent velocities attributed to P-waves (black vs. blue dots). Consistency along the FOC, i.e., along the depth, is strongly improved when using this narrow f-k filter. On the other hand, a significant decrease in SNR is observed locally around 450 m and 500 m for the 9 February and 22 April events. This decrease is associated with higher noise levels, as evidenced in Figure 4 from higher strain-rate amplitudes. 

With regard to the strain-rate amplitude (Figure 5c,d), the broad f-k filter isolating up-going wave fields does not substantially modify the maximum strain-rate compared with a standard BP filter (blue vs. red dots). This is less true for the shallowest SPs, nearer to the surface, on which significant down-going waves are recorded. The narrow f-k filter dedicated to the P-wave, however, decreases the maximum strain-rate amplitude (black vs. blue dots). This is particularly significant for the “weak” April event.

In view of the impact of this narrow f-k filter on SNR and signal coherency (panels (a) and (b)) and SR amplitudes (panels (c) and (d)), we solely use it to trigger data and determine onset, times of P- and S-waves, but not when signal amplitude should be measured.

### 4.2. Event Location

DAS provides unidirectional measurements, i.e., along the optical fiber. According to the setup at SLS, the DAS may be considered an antenna recording vertical strain along the first section of TH3. No polarization of the incoming waves can be estimated. This configuration makes it possible to estimate the depth and offset of the seismic source relative to the DAS antenna. However, a comprehensive analysis of hypocenters requires complementary observation stations, such as surface seismometers or DAS from other wells. Hence, the hypocenters presented in this section are computed using P- and S-wave onset times observed on all available stations of the seismometer network and DAS SP in TH3. The latter are uniformly sampled along the FOC from 150 m (bgl) to the lower end of the cable to ensure optimal conditions for the automatic picking of onset times (see Section 4.1).

Figure 6 shows the location of the 22 April event. It is derived by combining the onset times measured on 55 DAS SPs sampled every 10 m along TH3 and those obtained at the SYBAD station, the only seismometer to record this event. The cumulative probability around the most likely hypocenter is color-coded and clipped at 68% to show the extent of the ±1 standard deviation confidence interval. The result is projected on one horizontal and two vertical planes (panels (a–c)) and is visualized in 3D (panel (d)), highlighting the typically curved confidence ellipsoid that is oriented towards the DAS antenna and the SYBAD station.

The location of the 22 April hypocenter is detailed in Table 1. It is consistent with the initial estimate proposed by [17], which was based on Wadati diagrams [49] and supported an event at shallow depth, within the sedimentary cover. Table 1 also includes the event origin time, the time residual, and the hypocenter uncertainty. The latter is quantified by the length of the major axis of the confidence ellipsoid, i.e., ±222 m. Table 1 also includes the results obtained by changing the number of input DAS observations in the location procedure. This aspect will be discussed in Section 5.1. 

The stronger 9 February event could be located using observations from all seismometers of the local network and 55 DAS SP. The associated location is detailed in Table 1, and alternative results are computed by varying the number of DAS observations. The hypocenter suggests that the event occurred in the granitic basement with an uncertainty of ±382 m on the measured depth. The epicenter, which is located about 10 km from the TH3 wellhead, is significantly more distant than the 22 April event. It is also close to the south-east boundary of the local monitoring network (Figure 1).

### 4.3. Strain-Rate to Acceleration

Figure 7 depicts the conversion of DAS strain-rate data to acceleration for the 9 February (panel (a)) and 22 April (panel (b)) events. Particular attention is paid to the trace recorded at 180 m (solid red line), as it will support the comparison to the SYBAD seismogram in Section 5.2. The top panels show all traces for the initial 300 m in TH3 and highlight the 100-m-long slant-stack window (separated by dashed red lines). The bottom panels show the semblance matrix computed from the slant-stack procedure detailed in Section 3.2.2. Both matrices highlight clear variations associated with different phase arrivals in the slowness-time domain. The slowness time series (red line in the semblance matrix) is used in Equation (1) to convert strain-rate to acceleration.

### 4.4. Seismic Source Parameters

Figure 8 and Figure 9 show the seismic source parameters for the 9 February and 22 April events, respectively. They are computed with the converted DAS traces and event locations, as described in Section 3.2.3. For both events, panel (a) compares the amplitude spectrum (solid line) observed for the SP at 180 m to the best-fitting synthetic spectrum (dotted line) described by Equation (2). The fitting procedure allows estimating the corner frequency *f*_0_ and amplitude plateau Ω_0_. Equations (3)–(5) are then applied to derive subsequent source parameters. Panels (b) and (c) illustrate the spatial variability along the FOC, with the parameters being estimated every 12.5 m. Average values are also detailed in Table 2 for both events. 

To compare the parameters estimated from different measurement types, we apply the same procedure to data recorded by surface and downhole seismometers. Since the SYBAD station is the closest to SLS and recorded both events, it is taken as a reference station. Table 2 includes the source parameters computed from this station. Figure 8 and Figure 9 show the ratio of all estimated seismic moments and stress drops with the corresponding SYBAD reference values.

For both events, the DAS-based *M*_0_ estimates are in good agreement with the reference value (Figure 8b and Figure 9b). This is evidenced by the average ratios, which are μ = 1.1 ± 0.21 and μ = 0.97 ± 0.36 for the 9 February and the 22 April events, respectively. The relatively small standard deviations show the consistency of the measured values along the TH3 fiber. This can greatly contribute to a more comprehensive and robust statistical analysis of the seismic moment compared with sparsely distributed stations. Although the average stress drops are also relatively close to the reference value, their dispersion along the optical fiber is larger (Figure 8c and Figure 9c). For the stress drop ratios, we calculate an average value of 1.05 ± 0.64 and 0.83 ± 0.53 for the 9 February and the 22 April events, respectively. The comparison with the SYBAD station demonstrates that the average DAS-based corner frequencies and stress drops are consistent with those estimated from the adjacent broadband seismometers.

We use the 9 February event to compare the variability of the source parameters across the seismometer network and the optical fiber. Figure 8d,e shows the seismic moment and stress drop ratios computed between SYBAD and other seismometers within the network. The average ratio is μ = 0.87 ± 0.39 for the scalar moment and μ = 0.66 ± 0.55 for the stress drop. These results point out the variability of values observed on the network of surface and borehole seismometers, which may be larger than that observed with DAS along the optical fiber.

The comparison of the source parameters obtained from DAS and seismometers indicates that the converted DAS strain-rate gives access to robust magnitude estimates. DAS, with its intrinsic large number of SP, enhances the statistical robustness of the source parameter determination. The consistency between DAS and seismometer estimates suggests that the phase and amplitude of DAS acceleration traces faithfully represent ground motion across the studied frequency range. This topic is further discussed in Section 5.2.

## 5. Discussion

This section addresses the challenge described in the “Introduction” concerning the influence of DAS SPs density on event location (Section 5.1) and on computation times (Section 5.3). In addition, we provide further confirmation that the converted DAS measurements accurately reflect actual ground motion (Section 5.2).

### 5.1. Event Location

DAS data collected from the TH3 well at SLS are insufficient for resolving the events hypocenters, which requires the inclusion of additional observation points. The 9 February event could be located using the local seismometer stations since its magnitude *M*_W_1.7 was higher than the magnitude of completeness of the network. On the other hand, the 22 April event was visible solely on the borehole seismometer at SYBAD, owing to a lower magnitude of *M*_W_-0.1. Given the high density of SP in TH3, a question arises regarding the relative weight of DAS observations in resolving the hypocenters of seismic events. We address this question by analyzing the effect of a varying number of DAS observations on the location results. The results are detailed in Table 1 and illustrated in Figure 10.

First, we used the highest number of observations while ensuring their independence. Since the acquisition gauge length is set to 10 m, it defines the smallest offset between two successive and independent observations. Hence, to cover the 150–700 m depth range, the largest number of SP along the TH3 FOC is 55 s; we selected the same number of DAS SP as the number of seismometers. For the 9 February event, the number of DAS SPs was accordingly reduced to 11, with one observation taken every 50 m. For 22 April, only one SP is selected and taken at the same depth as the SYBAD seismometer, 180 m bgl. Finally, the 9 February event was also located using only the seismometer network, and the 22 April event had an intermediate number of six SP sampled every 100 m.

Figure 10 shows the extent of the 68% confidence interval in each case. As observed and detailed in Table 1, the location uncertainties are decreasing with the increasing number of DAS SP. This observation can be attributed to the comparable picking uncertainty assigned to each DAS onset time. However, the hypocenters do not fit. The relative spatial shifts observed in Figure 10 are counterbalanced by an adjustment of the origin time of the event, the latter being a central part of the inverse problem. The 22 April event is particularly illustrative of this scenario: the addition of SP decreases the location uncertainty, the time residuals, and the origin time but deepens the hypocenter (Table 1). For the 9 February event, accounting for DAS SP in the location procedure moves the hypocenter 2.5 km northward, approximately 350 m westward, and at greater depths. These shifts in hypocenter, influenced by the quantity and positioning of the observational data, suggest the existence of inaccuracies in the location results and indicate systematic errors. These bias stem from discrepancies between the velocity model and the actual geological conditions. Consequently, increasing the number of DAS SP in the event location increases the relative weight of the velocity model between the FOC and the hypocenter.

Moreover, as seen in Figure 10, the variations in hypocenter are not covered by any of the 68% confidence intervals. This illustrates that relatively small uncertainties do not guarantee that the hypocenter is correct. Indeed, the imprecision (or uncertainty) in location depends on the picking uncertainties addressed by the location inverse problem, while inaccuracy in location arises from the disparity between the velocity model and the actual geological conditions (see Section 3.2.1). Similar issues have been previously modelled and discussed [2,50,51]. They could potentially result in incorrect mapping of subsurface faults and fractures inferred from seismic activity.

This analysis emphasizes the importance of having P- and S-wave velocity models that accurately reflect the true geological conditions between the seismic source and the receivers, especially as the number of DAS sensing point increases. In this study, these velocity models are constructed using well data, including zero-offset and offset VSPs as well as sonic logs, following the procedure outlined in Appendix A (Text A1). While the models’ layer interfaces are mapped throughout the study region from 3D- or 2D-seismic migrations, interpolation between the wells with existing observations are necessary to cover the volume of interest. Increasing the reliability of these models deeper than the wells is particularly difficult, since direct observations are generally not available at depths. Yet, this knowledge is necessary to locate seismic events occurring in or below the geothermal reservoir. To improve the velocity models in this volume, passive seismic tomography could be used (e.g., [52]). Nevertheless, this requires sufficient and spatially distributed seismicity, which is limited in our specific geothermal context. Another approach could be to calibrate the models using active sources deployed in wells. For instance, perforation shots of the production tubing. (e.g., Ref. [53]) are frequently used for the seismic monitoring of hydraulic fracturing jobs, in enhanced oil and gas recovery. Such a procedure is depending on the production completion scheme and may not be applied. However, the concept of an active source positioned deep in a well (reaching the reservoir), and whose signal could be captured by the existing seismic monitoring network, would be valuable for velocity models calibration.

### 5.2. Assessment of DAS-Based Acceleration

DAS strain-rate measurements may be significantly influenced by the intrinsic characteristics of the optical fiber cable, the installation configuration, laser signal attenuation, and spatial resolution, which can introduce biases in seismic applications requiring accurate ground motion measurements. Previous studies emphasized the influence of several factors associated with the manufacturing and installation of FOC on DAS data, such as the fiber packing in the cable, its outer coating, or the way it is coupled to the surrounding medium [54]. Similar issues were investigated using a physical model of a DAS cable, based on actual material properties [16]. In the present study, the FOC is cemented behind casing and the fiber is a tight optical fiber, which is known to be favorable for DAS data quality. Recent studies investigated also the DAS instrument response in different settings and over a large range of frequencies [11,25]. They demonstrated that DAS could be used in seismic applications requiring a quantitative evaluation of the waveforms. However, benchmarking the ground motions obtained from DAS with standard seismometers remains a key aspect to ensure accuracy and reliability of DAS data and associated processing results. In this section, we propose to compare the DAS-derived acceleration at SLS with that calculated from the SYBAD borehole seismometer located about 1 km away.

Figure 11 shows the DAS waveforms recorded at 180 m bgl (i.e., the installation depth of the SYBAD station) and the SYBAD waveforms for the 9 February and the 22 April event (panel (a) and (b), respectively). The original records have been converted to acceleration, filtered between 5 and 40 Hz and time synchronized by cross-correlation. To quantitatively compare these traces, a goodness-of-fit (GOF) test is applied as suggested by [55,56]. The test compares both signals in the time and the frequency domains. The continuous (Morlet) wavelet transform of each trace allows the calculation of local envelope and phase differences. In the temporal domain (horizontal axes of subplots in Figure 11), respectively frequency domain (vertical axes of subplots in Figure 11), TEG and TPG, respectively FEG and FPG, measure the envelope and phase GOF. These GOF values can range from zero, corresponding to a ±π difference in phase and ±∞ difference in envelope, to 100, indicating a perfect match in phase or envelope.

The phase information of DAS and seismometer waveforms are particularly consistent, as suggested by the average GOF values in frequency and amplitude (TPG, FPG), which are mostly higher than 80%. For the envelope information, this consistency is also observed over time and frequency (TEG, FEG). More particularly, the DAS measurements (Figure 11, in red) accurately capture the initial P-wave arrival observed at the SYBAD station (Figure 11, in black). For the 9 February event, the difference between the waveforms is more important for the direct S-wave. For the 22 April event, Figure 4 and Figure 11 show that the direct S-wave is faintly visible below 200 m. Although the sensors are not co-located, the observed similarities emphasize the robustness of the DAS-data conversion procedure. It also indicates that both recording types can be jointly used in the seismic monitoring workflow to assess seismic activity quantitatively.

The procedure proposed for DAS data conversion is relatively computationally intensive since it requires the calculation of the semblance matrix across both spatial and temporal dimensions for each trace considered. For this reason, we tested a simplification of the slowness time series, which consisted of assigning a constant slowness of 0.4 km/s before the S-wave onset time and a constant slowness of 1.1 km/s after it (see Figure A4, Figure A5 and Tabla A1 in Appendix A). The resulting waveform was compared with the one obtained following the original slant-stack procedure (Figure 11, red curve) and with the SYBAD acceleration trace (black curve). The GOF measured with the SYBAD waveform is higher when applying the procedure suggested in Section 3.2.2, and we observe an amplification of the noise in the case of the simplified slowness time series. Hence, although expensive in terms of computational resources, the full slant-stack method is more reliable. Furthermore, it has the advantage of being based on observational data and does not depend on a pre-established velocity model. In a more complex scenario, for instance, an FOC in a deviated well or spanning multiple geological units, this method could account for local variation of the apparent velocities along the optical fiber.

On a long-term basis, it would be important to periodically test the quality of the optical signal to guarantee a correct SR to acceleration conversion, especially if no nearby 3C seismometer is available any longer. The use of an optical time-domain reflectometer (OTDR) would be appropriate for a quality check of the optical fiber over time (e.g., the appearance of reflection points and signal losses at the connectors).

### 5.3. Implications for Seismic Monitoring Systems

Seismic monitoring of geothermal reservoirs, and more broadly, geo-reservoirs, requires an infrastructure capable of handling the flow of data and its storage. With this regard, the implemented data management scheme facilitated the secure and seamless transfer of DAS data from the interrogator to the cloud infrastructure where they are stored. DAS in TH3 generated 3.5 TB/month, but the cloud solutions used in this study provide scalable resources [57,58] that could accommodate larger amounts of data. Hence, complementary monitoring components, such as additional DAS antennas, seismometers, and field production parameters, can be later integrated, further contributing to field monitoring. In addition, the data access policy we applied defined user-dependent rights and hierarchical storage levels. Access tiers, which are categorized as hot, cool, or archive, define different performance requirements. The hot tier was, in particular, selected for high-performance data processing on cloud workstations, including event detection and picking. Local workstations were used for event characterization, including event location, because the company’s cloud solution prevented us from compiling programs, such as the location software NonLinLoc.

Although the waveform conversion and source characterization tasks were carried out on the local workstation, a performance comparison was conducted with cloud-based workstations to evaluate their capability to deliver results with lower latency. In both cases, multithreading was implemented to optimize the processing of the waveforms recorded by the SP. The analysis shows that the frequency with which results are delivered can be halved, from 28 to 13 s per SP, depending on the number and frequency of the processors (see Figure A6). With this regard, a cloud workstation gives access to scalable and high-end processing resources, which is particularly advantageous for the resource-intensive conversion from SR to acceleration (see Section 5.2). Therefore, executing the complete processing workflow on the cloud has the potential to deliver results faster. To this end, an alternative solution for event location has to be investigated, which is the scope of future work.

## 6. Conclusions

This study demonstrates that borehole Distributed Acoustic Sensing can be used as an integral part of the seismic monitoring of the SLS geothermal field. The infrastructure implemented could efficiently manage the generated amount of DAS data and automatically provide valuable observations to characterize the seismic activity. As a result, a catalog containing event origin time, hypocenter, uncertainties, seismic moment (and moment magnitude), and stress release can be delivered automatically.

Cloud-based services used by the geothermal field operator were leveraged. They handled the secure storage of data and a major part of its processing continuously and automatically. In this study, automatic event detection and phase picking were carried out on the cloud workstations, and the seismic processing of the detected events was conducted on local workstations. Tests showed that it is possible and worth migrating the latter to the cloud in order to benefit from scalable and performant computing resources. Further work is, however, necessary to migrate the location task, especially in 3D velocity models.

The infrastructure and the data processing workflow were illustrated with two noteworthy events, providing proof of the viability of the concept for long-term seismic monitoring. We showed that the DAS strain-rate could be reliably converted to acceleration from the apparent wavetrain slowness, which allowed the application of state-of-the-art seismic processing to the DAS-based data. We also demonstrated that DAS-based phase and amplitude analyses allow for a reliable assessment of the seismic source parameters. Consequently, converted DAS data were combined with data recorded by the local seismological network. This confirms the applicability of hybrid networks with a DAS component.

Furthermore, borehole DAS contributed to increasing the sensitivity of the existing monitoring network despite being deployed in the city of Munich at the Schäftlarntraße geothermal plant. DAS on the cemented fiber optic cable detected the *M*_W_-0.1 22 April event, which might otherwise have gone unnoticed. The application of array-processing techniques, in particular the f-k filtering, significantly improved the signal-to-noise ratio. The high density of sensing points also led to more certain hypocenter determinations. Although uncertainties may appear small, failure to account for potential discrepancies between the P- and S-wave velocity models and reality can result in systematic location errors. Hence, developing accurate velocity models remains a crucial aspect of seismic monitoring, albeit not the sensing capabilities provided by DAS.

A short-term perspective is to implement the whole processing workflow in the cloud and optimize it for rapid delivery of the results. Progress towards real-time seismic monitoring and integration of the results to mitigate seismic hazards is paramount for geothermal field operators. The scalability of the proposed cloud infrastructure and the automatic processing procedure give additional perspectives to this work. It could accommodate larger monitoring networks and manage additional DAS components or be replicated to monitor the numerous geothermal fields under development in the Munich area.

## Figures and Tables

**Figure 1 sensors-24-03061-f001:**
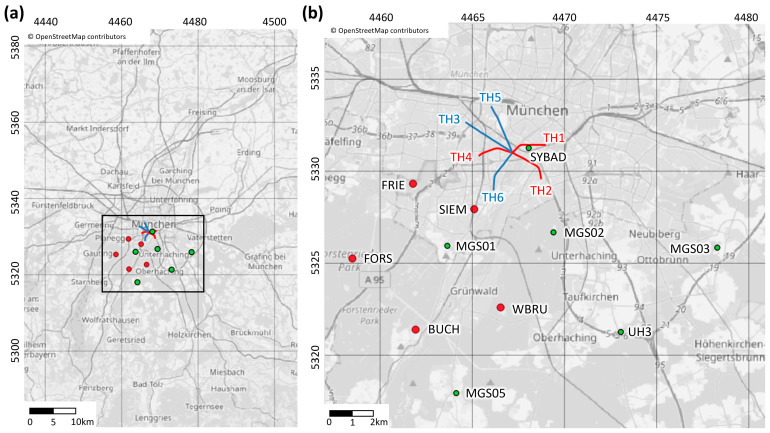
Overview of the study area (panel (**a**)) and zoom around the zone of interest in the south of Munich (panel (**b**)). The Gauss–Kruger 4 coordinate system (EPSG 31468) is used (in km). The trajectories of the six wells of the SLS geothermal site are projected. The red and blue lines show production and injection wells, respectively. The dots show the location of the seismic stations used in this study, including two borehole stations (SIEM, SYBAD). All other stations are installed on the surface. These seismic stations are in a radius of 15 km around SLS and continuously monitor the area.

**Figure 2 sensors-24-03061-f002:**
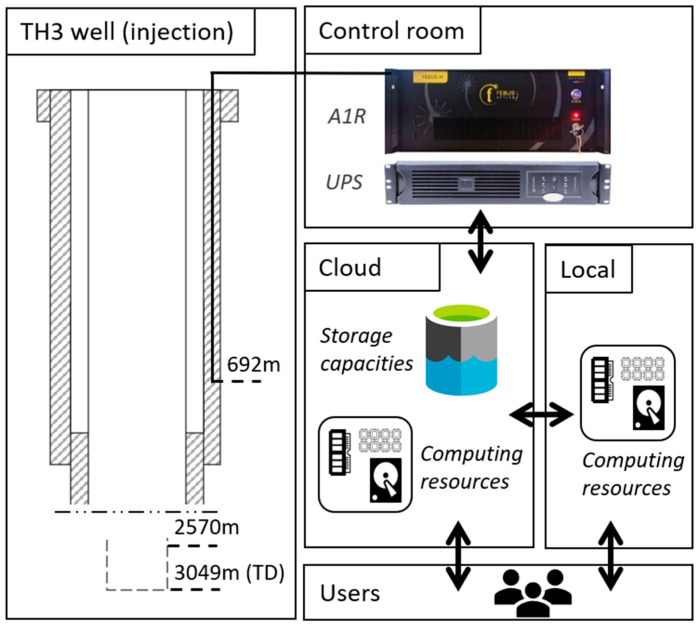
Overview of the DAS monitoring system using the TH3 FOC at SLS. The left-hand side shows the completion of the injection well TH3 along its first 900 m (top) and the open-hole section until the well total depth (bottom). The TH3 FOC is cemented behind the casing. The micro-bend located at 692 m connects two individual fibers of the unique FOC, which creates a so-called U-loop and doubles the number of sensing points along TH3. One end of the TH3 fiber is connected to the Febus A1-R interrogator located on-site in the control room. The sketch on the right-hand side shows the infrastructure set up between the DAS interrogator, the cloud storage resources, the cloud and local processing resources, and the users of the infrastructure. The arrows describe the interactions between each component.

**Figure 3 sensors-24-03061-f003:**
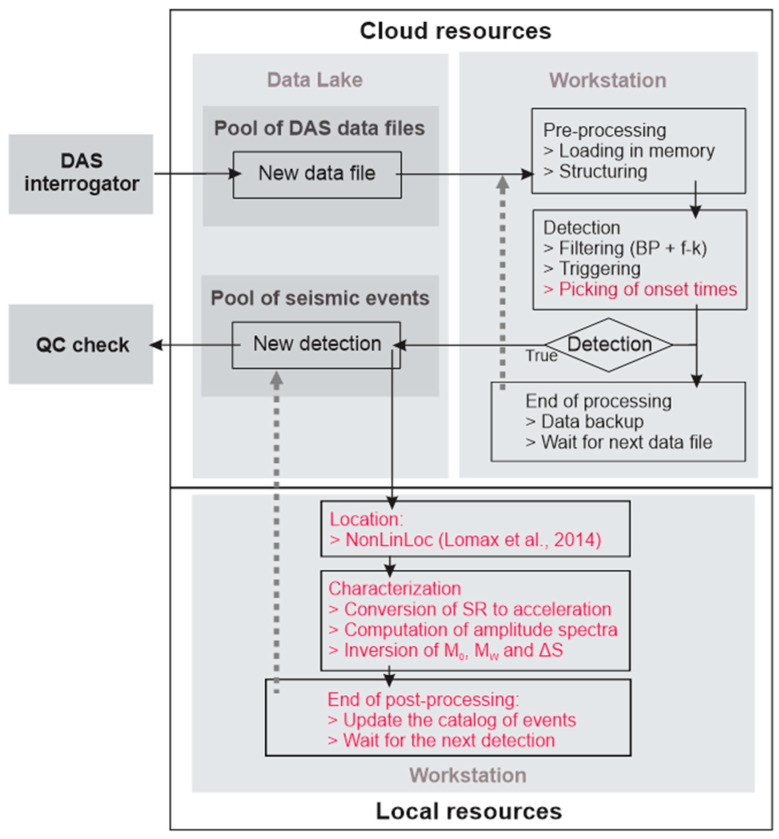
Flowchart depicting the storage and processing of DAS data files, starting from their creation by the interrogator to the validation of processing results. The chart distinguishes between the resources located in the cloud (top) and the local resources (bottom). The computing resources are of two kinds. Cloud-based workstations are used for data pre-processing and continuous detection of new seismic events. Local workstations are used for seismic source characterization. The cloud storage environment (or Data Lake) is at the interface between these processing resources.

**Figure 4 sensors-24-03061-f004:**
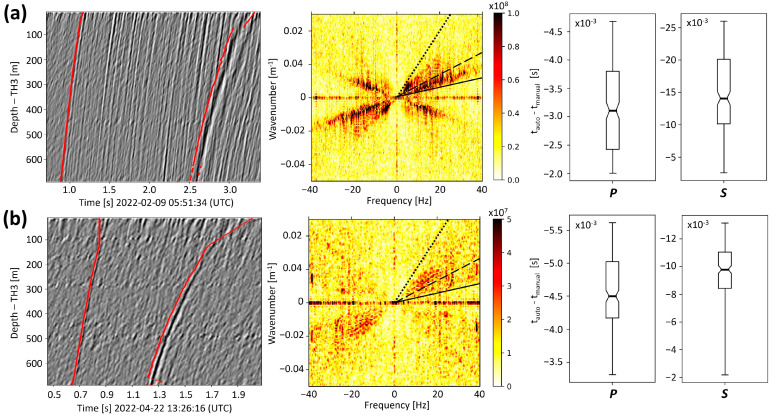
Panel (**a**) corresponds to the 9 February event, and panel (**b**) to the 22 April event. The left panels show the BP and f-k filtered SR datasets and the signature of the first P- and first S-wave arrivals. The red solid curves show the results of the automated onset time picking of the P- and the S-phases. The middle panels show the unfiltered SR datasets in the f-k domain. The solid, dashed, and dotted lines correspond to apparent velocities of 3500, 1600, and 500 m/s, respectively. The box plots on the right panels show the differences between manually and automatically picked onset times for the first P- and S-waves and below 150 m. The box extends from the first to the third quartile values of the data, with a line at the median. The whiskers depict the range of values.

**Figure 5 sensors-24-03061-f005:**
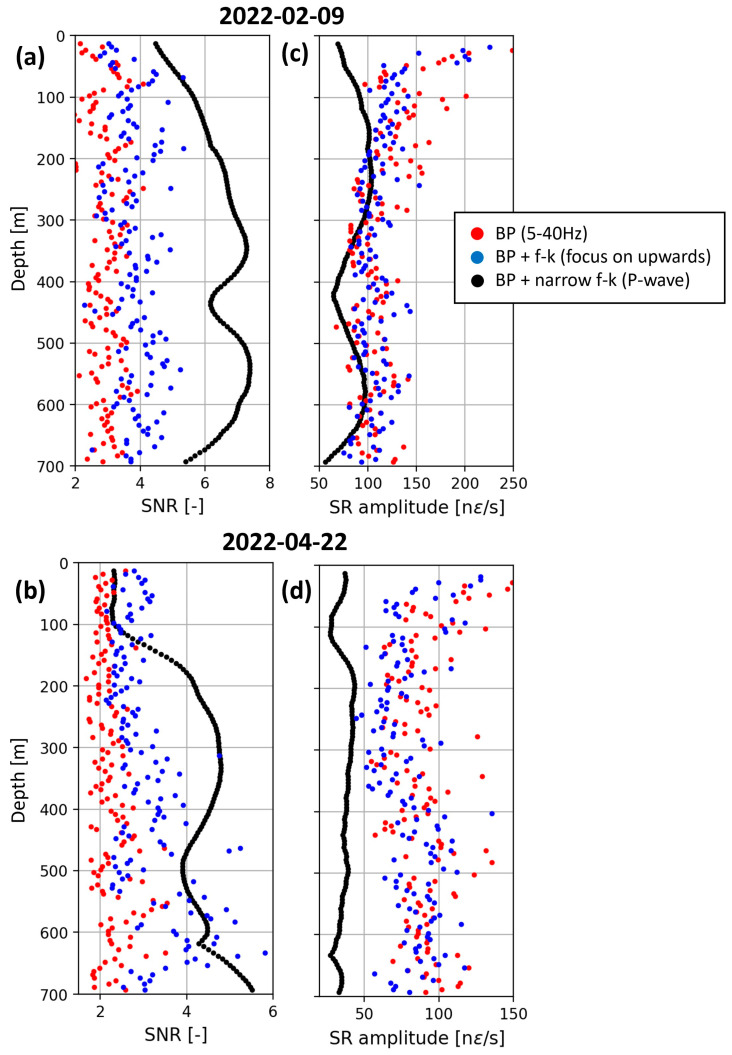
Evolution with depth of the SNR ([-]—panels (**a**,**b**)) and maximum strain-rate amplitude (in nanostrain per second, panels (**c**,**d**)) of the first arrivals recorded for the 9 February (top) and 22 April (bottom) events. Different filtering strategies are applied. The datasets are only BP filtered between 5 and 40 Hz (red dots), or BP and f-k filtered by selecting the f-k domain associated with upwards going wave fields (blue dots), or BP and f-k filtered by selecting the f-k domain associated with the P-waves propagating along the FOC (black dots).

**Figure 6 sensors-24-03061-f006:**
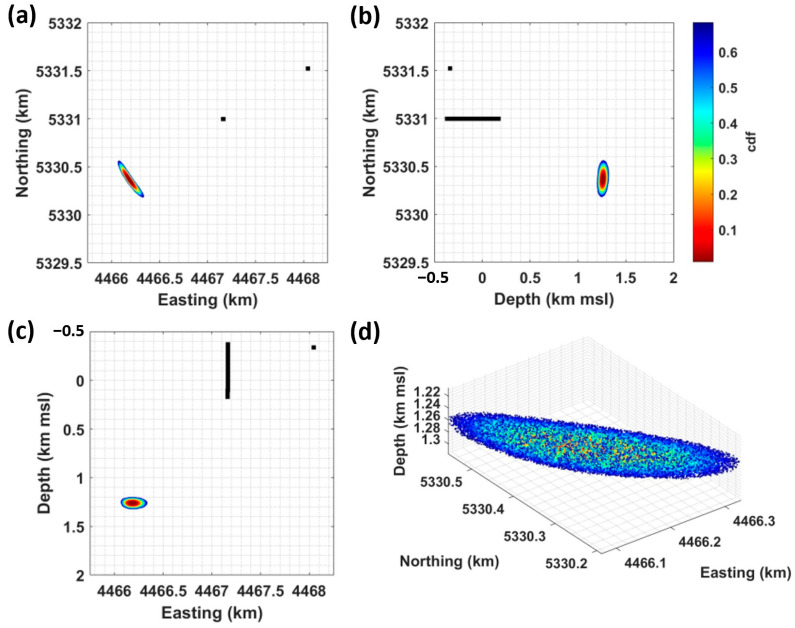
Location of the 22 April event obtained from SYBAD and 55 DAS SP. The cumulative probability around the most probable hypocenter (dark red) is color-coded up to 68% (dark blue) to show the extent of ±1 standard deviation confidence interval. The black dots show the position of the sensors used in the location procedure. The panels (**a**–**c**) present projections of the result on different planes, and panel (**d**) provides a 3D view of the result. The coordinate system is Gauss-Krüger 4 (EPSG 31468), and depths are referenced to mean sea level (msl).

**Figure 7 sensors-24-03061-f007:**
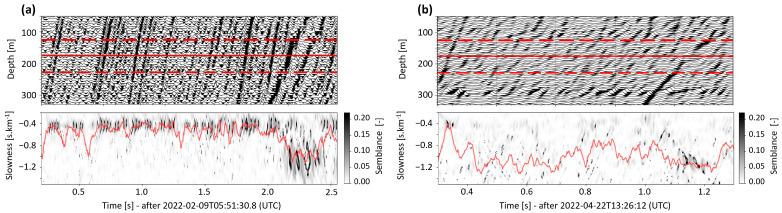
Time-variation of the slowness of the 9 February (panel (**a**)) and 22 April (panel (**b**)) seismic events. The top panels show the data subset around the trace under consideration, here at 180 m (solid red line), and the sub-domain used to calculate the semblance matrix (delimited by dashed red lines). The bottom panels show the semblance matrix, i.e., the temporarily varying semblance values (grayscale) calculated for a range of slowness (vertical axis). The time-averaged maximum semblance value (red line) defines the slowness of the time-series used for data conversion. See Section 3.2.2 for calculation details.

**Figure 8 sensors-24-03061-f008:**
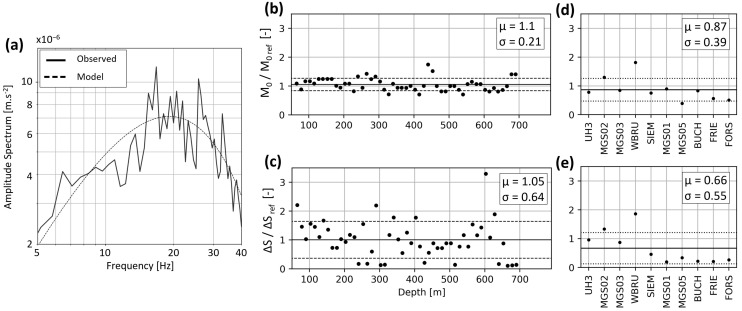
Evaluation of the source parameters for the *M*_W_1.5 9 February event and comparison between DAS- and seismometer-based estimates. Panel (**a**) shows the amplitude spectrum observed at 180 m (solid line) and the best-fitting synthetic spectrum (dashed line) in the 5–40 Hz bandwidth. Panel (**b**,**c**) shows the ratios of seismic moments and stress drops obtained from DAS along the FOC with the reference value computed at SYBAD. Panels (**d**,**e**) show the ratios of seismic moments and stress drops obtained at 10 local seismometer stations with the reference value at SYBAD. In panels (**b**–**e**), the average value, μ (solid line), and standard deviation, σ (dashed lines), are displayed, and the values are given in the inlet.

**Figure 9 sensors-24-03061-f009:**
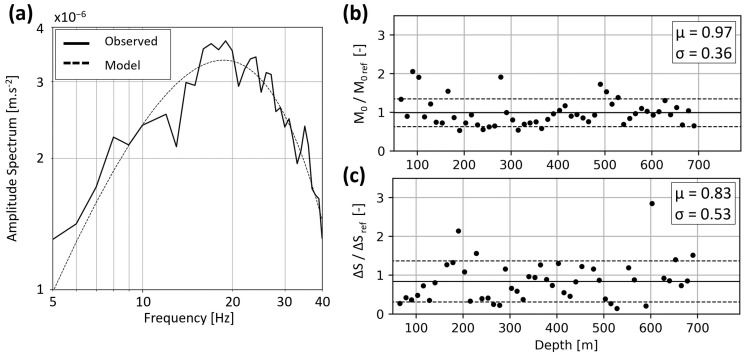
Same as Figure 8 for the 22 April event. Panel (**a**) shows the amplitude spectrum observed at 180 m (solid line) and the best-fitting synthetic spectrum (dashed line) in the 5–40 Hz bandwidth. Since this event was only visible on the SYBAD seismometer, we consider only the seismic moment and stress drop ratios between the DAS estimates and the SYBAD reference values (panels (**b**) and (**c**), respectively).

**Figure 10 sensors-24-03061-f010:**
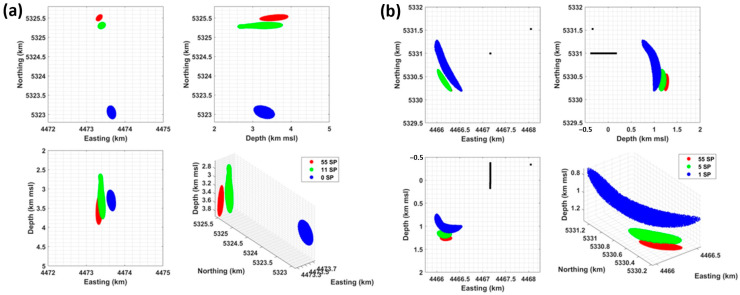
Location results with different combinations of DAS SP for the 9 February (panel (**a**)) and 22 April (panel (**b**)) events. The 68% confidence ellipsoids computed for each combination are projected on three orthogonal planes and shown in 3D using different colors. The number of DAS SP used in each case is indicated in the legend. The coordinates system is Gauss-Krüger 4 (EPSG 31468), and depths are referenced to mean sea level (msl).

**Figure 11 sensors-24-03061-f011:**
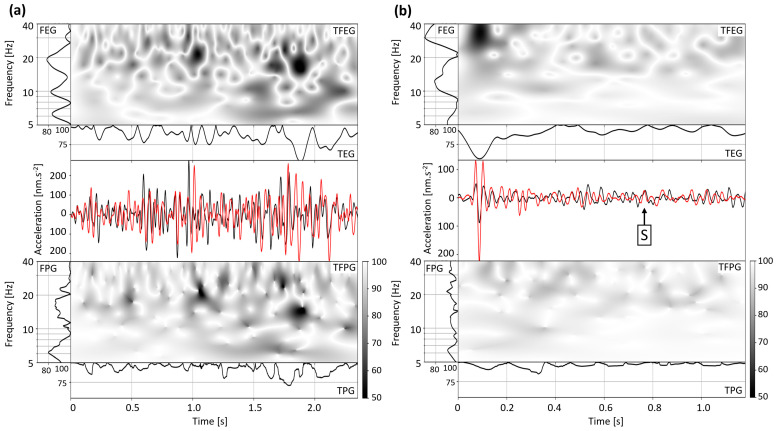
Panel (**a**) focuses on the 9 February event, and panel (**b**) on the 22 April event. The middle rows compare the vertical channel of the SYBAD station (black curve), and the DAS waveform at 180 m in TH3 (red curve) once converted to acceleration. The top rows show the normalized goodness-of-fit (GOF) of the waveform envelopes in the time-frequency domain (TFEG) and its average over frequencies (FEG) and over time (TEG). The content of the bottom rows is similar to the top rows, but the GOF is calculated on the waveform phases (TFPG, FPG, TPG).

**Table 1 sensors-24-03061-t001:** Location results for the 9 February (top) and 22 April (bottom) seismic events, using a variable number of DAS sensing points. The table gives details on the number and spacing of DAS SP selected for each location result (first two columns), the hypocenter coordinates and origin time (*t*_0_). The column “RMS” gives the root-mean-square of the time residuals obtained at the hypocenter. The last column indicates the length of the semi-major axis of 68% confidence ellipsoid. The location results in bold are obtained with 55 DAS SP and serve as a reference point for the discussion in Section 5.1.

Event	Number of DAS SP	DAS Spatial Sampling	Easting [km GK4]	Northing [km GK4]	Depth [m msl]	*t*_0_ (UTC)	RMS [s]	Len [m]
9 February	**55**	**10 m**	**4473.33**	**5325.51**	**3570**	**2022.02.09T05:51:29.100**	**0.041**	**382**
	11	50 m	4473.39	5325.31	3320	2022.02.09T05:51:29.088	0.086	546
	0	-	4473.65	5323.06	3287	2022.02.09T05:51:29.161	0.078	288
22 April	**55**	**10 m**	**4466.19**	**5330.36**	**1260**	**2022.04.22T13:26:11.664**	**0.016**	**222**
	6	100 m	4466.14	5330.42	1177	2022.04.22T13:26:11.682	0.038	277
	1	-	4466.20	5330.66	1043	2022.04.22T13:26:11.772	0.068	649

**Table 2 sensors-24-03061-t002:** Seismic source parameters of the 9 February and 22 April events. The corner frequency is noted as *f*_0_, the low-frequency spectrum plateau Ω_0_, the scalar seismic moment *M*_0_, the moment magnitude *M*_W_, and the stress drop ΔS. The first five columns show the average parameters calculated from DAS. The last three columns *M*_0,ref_, *M*_W,ref,_ and ∆S*_ref_*, are the reference values measured at the SYBAD seismometer.

Event	*<f*_0_> (Hz)	<Ω_0_> (m.s^−2^.Hz^−1^)	<*M*_0_> (N.m)	<*M_W_*>	<∆S> (Pa)	*M*_0, ref_ (N.m)	*M_W, ref_*	∆S*_ref_* (Pa)
9 February	20	3.7 × 10^−9^	5.8 × 10^11^	1.7	1.2 × 10^7^	5.4 × 10^11^	1.7	1.1 × 10^7^
22 April	25	1.9 × 10^−9^	1.1 × 10^9^	−0.1	3.4 × 10^4^	1.1 × 10^9^	−0.1	4.2 × 10^4^

## Data Availability

The DAS strain-rate data of the 9 February and 22 April events and the scripts used on the workstations are openly available from the KIT Open Access repository at DOI: 10.35097/pxwZMAGAgqMCbRka (https://dx.doi.org/10.35097/pxwZMAGAgqMCbRka) (accessed on 2 May 2024).

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
