# Peer review of "Seismic Monitoring of a Deep Geothermal Field in Munich (Germany) Using Borehole Distributed Acoustic Sensing"

_sensors, 2024, doi:10.3390/s24103061_

Round 1

Reviewer 1 Report

Comments and Suggestions for Authors

Thank you for your submission on using borehole Distributed Acoustic Sensing (DAS) for seismic monitoring in the context of geothermal energy exploitation in Munich. I appreciate the comprehensive approach taken in your study and the detailed workflow you have introduced for seismic event detection and characterization. The validation against traditional seismometers adds credibility to your approach. Below are some comments and suggestions that could help improve your paper:

Highlights:

- The study applies a consolidated workflow for processing DAS data, from seismic event detection to the determination of seismic parameters.

- A 6-month monitoring period showcases the capabilities of the proposed system, validating the results against traditional seismometers.

- The paper demonstrates the potential of hybrid seismic networks, combining surface-based seismometers and borehole DAS, with scalability as geothermal operations expand.

-The study confirms that the high-density array configuration of borehole DAS is particularly beneficial in urban and operational settings.

Areas for Improvement:

- Reliance on Seismic Velocity Model: The abstract highlights the importance of a realistic seismic velocity model. However, it could offer more insights into the impact on the workflow when seismic velocity is wrong. 

- The authors could talk about the challenges and best practices for obtaining velocity models, especially in urban areas. 

- Quality control: The study could emphasize on how to keep quality control of data processing, specially regarding the mentioned aspects, such as  the intrinsic characteristics of the optical fiber cable, the installation configuration, laser signal attenuation, and spatial resolution.

- The validation against traditional seismometers adds credibility to your approach, but I would suggest the authors to expand the discussion on how the goodness of fit is measured.

Conclusion:
Overall, I find your paper to be a valuable contribution to the field of geothermal seismic monitoring. Addressing the above comments, and the below points, would strengthen your work and improve its impact on the scientific community. I look forward to seeing your revisions and believe that your research will be beneficial to further exploration in this domain.

 Points:

l. 120 - There is an extra dot.

l. 161-162  “The location of the study site, in the city of Munich, has a significant impact on seismic noise levels due to anthropogenic activities.” The location has a significant impact? 

Figure 7 - Semblance is not normalized? Why does the color scale goes up to 2?

Figure 7b - Include y-axis label

l.675 - Could you expand your comments about the goodness of fit. Why 5 is considered a good fit?

l.786 - Your repository link is broke

Thank you for your contribution and I look forward to read the final publication.

Author Response

Thank you for your submission on using borehole Distributed Acoustic Sensing (DAS) for seismic monitoring in the context of geothermal energy exploitation in Munich. I appreciate the comprehensive approach taken in your study and the detailed workflow you have introduced for seismic event detection and characterization. The validation against traditional seismometers adds credibility to your approach. Below are some comments and suggestions that could help improve your paper:

Thank you for your positive comments and constructive recommendations. We believe these will help to improve the overall quality of the manuscript. We have taken into consideration each of the comments made, addressing areas of improvements or minor issues. We provide a response to each of them below (in red). We focussed especially on the improvement of the description of the applied goodness of fit criteria and on adding details about the velocity model and ways to improve its calibration.

Highlights:

- The study applies a consolidated workflow for processing DAS data, from seismic event detection to the determination of seismic parameters.

- A 6-month monitoring period showcases the capabilities of the proposed system, validating the results against traditional seismometers.

- The paper demonstrates the potential of hybrid seismic networks, combining surface-based seismometers and borehole DAS, with scalability as geothermal operations expand.

-The study confirms that the high-density array configuration of borehole DAS is particularly beneficial in urban and operational settings.

Areas for Improvement:

- Reliance on Seismic Velocity Model: The abstract highlights the importance of a realistic seismic velocity model. However, it could offer more insights into the impact on the workflow when seismic velocity is wrong.

We modified the sentence in the abstract to be more specific: “The study stresses, however, that realistic prior knowledge of the seismic velocity model remain essential to prevent the large number of DAS sensing points from biasing results and interpretation”.

We also outlined that the detection and automatic picking processes are independent of the velocity model in subsection 3.1 : “After filtering, the detection of events is based on a network coincidence approach imple-mented in the Obspy library [33] and using a recursive STA/LTA algorithm [38,39]. This procedure is independent from the velocity model.”

- The authors could talk about the challenges and best practices for obtaining velocity models, especially in urban areas.

In the discussion, we added new content to Section 5.1 (last paragraph) to discuss how the velocity model could be improved, using passive or active-sources based methods.

In our opinion, the problem is not so much the urban environment since VSP, 2D- or 3D-seismic campaigns can be run in cities, and this was already the case several times in Munich and its suburb. The challenge is rather to have a good knowledge of the velocities below the well, in the deep high velocity formations (e.g. the basement sitting below the reservoir in our case), in which the seismic waves would propagate. This specific aspect was outlined in the discussion.

- Quality control: The study could emphasize on how to keep quality control of data processing, specially regarding the mentioned aspects, such as the intrinsic characteristics of the optical fiber cable, the installation configuration, laser signal attenuation, and spatial resolution.

At the end of the subsection 5.2, we added the following paragraph: “On a long-term basis, it would be of importance to test periodically the quality of the optical signal to guarantee a correct SR to acceleration conversion, especially if no nearby 3C-seismometer is available any longer. The use of an optical time-domain reflectometer (OTDR) would be appropriate to check the quality of the optical fibre over time (e.g. appearance of reflection points, signal losses at the connectors).”

- The validation against traditional seismometers adds credibility to your approach, but I would suggest the authors to expand the discussion on how the goodness of fit is measured.

The test applied to compare the signals is using time frequency goodness of fit (GOF) criteria that are computed following the approach proposed by Kristekova et al. (2009). The latter are based on the misfit functions introduced in the original paper by Kristekova et al. (2006) (see Section 5.2).

We added additional details to Section 5.2 to better the description of the test and detail the steps involved in the computation of the criteria. In particular, the time-frequency GOF criteria are based on the complete time-frequency representation of the signals, which is obtained by continuous Morlet wavelet transform. Local time-frequency envelope and phase differences are computed from the time-frequency representation.

These criteria are suitable for comparing arbitrary time signals in their entire time-frequency complexity, as outlined by Kristekova et al. (2009).

Kristeková, M.; Kristek, J.; Moczo, P. Time-Frequency Misfit and Goodness-of-Fit Criteria for Quantitative Comparison of Time Signals. Geophysical Journal International 2009, 178, 813–825, doi:10.1111/j.1365-246X.2009.04177.x.

 Kristekova, M.; Kristek, J.; Moczo, P.; Day, S.M. Misfit Criteria for Quantitative Comparison of Seismograms. Bulletin of the Seismological Society of America 2006, 96, 1836–1850, doi:10.1785/0120060012

Conclusion:
Overall, I find your paper to be a valuable contribution to the field of geothermal seismic monitoring. Addressing the above comments, and the below points, would strengthen your work and improve its impact on the scientific community. I look forward to seeing your revisions and believe that your research will be beneficial to further exploration in this domain.

Points:

  1. 120 - There is an extra dot.

We removed the extra dot.

  1. 161-162  “The location of the study site, in the city of Munich, has a significant impact on seismic noise levels due to anthropogenic activities.” The location has a significant impact? 

We rephrased the sentence: “The location of the study site, in the city of Munich, implies a significant level of seismic noise, mainly attributed to anthropogenic activities.“

Figure 7 - Semblance is not normalized? Why does the color scale goes up to 2?

Thanks for outlining the mistake. We updated the scale in Figure 7 accordingly.

Figure 7b - Include y-axis label

We added the labels to figure 7b, on the Y-axis. In general, we updated all the figures to include the ones with sub-figures as a single picture.

 l.675 - Could you expand your comments about the goodness of fit. Why 5 is considered a good fit?

We removed Table 3 to give more emphasis to Figure 7, which describes the time-frequency variation of the GOF criteria and which was not sufficiently used in the previous version of the manuscript. We decided to avoid the reference to “good” fit. On the other hand, we added more details about the computation of the GOF and the interpretation of the associated scores. We scale this score out of 100 (instead of 10), which gives a more intuitive idea of the fit between the signals. In particular, phase and envelope similarity reach significant scores, above 80%, over frequency and time.

l.786 - Your repository link is broke

This is the link to the archive mentioned in the article:

https://bwsyncandshare.kit.edu/s/nidawMCigsXdp56

The link has been tested and provides access to the expected files

Thank you for your contribution and I look forward to read the final publication

Reviewer 2 Report

Comments and Suggestions for Authors

The authors have proposed a comprehensive workflow for automating the processing of DAS data acquired from a deep geothermal field. This presentation is interesting and should be useful for those working with DAS and geothermal monitoring. Generally, the paper is well written; however, I found it somewhat hard to follow at times. For example, in Figure A1, everything seems clear from the plots themselves but I am confused by the caption. Similar feelings arose with other figures as well.

Therefore, my suggestion would be for the authors to try making their paper more compact and neat while also being direct to their points made throughout it. 

Author Response

The authors have proposed a comprehensive workflow for automating the processing of DAS data acquired from a deep geothermal field. This presentation is interesting and should be useful for those working with DAS and geothermal monitoring. Generally, the paper is well written; however, I found it somewhat hard to follow at times. For example, in Figure A1, everything seems clear from the plots themselves but I am confused by the caption. Similar feelings arose with other figures as well. Therefore, my suggestion would be for the authors to try making their paper more compact and neat while also being direct to their points made throughout it.

>> Thank you for your review. We tried to shorten and clarify the manuscript. We focused on reducing wordiness and eliminating irrelevant parts of the text and figures, in order to make them more concise and focused.

For example:

  • We revised Figure A1 and included sub-figures for the correlation diagrams instead of using embedded figures, which may be confusing. Axis labels have been added. We modified the caption of the figure accordingly and we tried to be more precise in describing its contents.
  • We revised all figure captions with the objective to focus on the figure description, without adding elements about its analyses.
  • We removed the lithological unit of Figure 2 that was unused in the rest of the manuscript to simplify it and the caption associated.
  • We removed Table 3 in order to give further emphasis on Figure 11, describing the results of the good of fit test more extensively in both the time and frequency domain.

We hope that the additions and modifications brought to the manuscript will make the reading more pleasant and our message easier to understand. 

Reviewer 3 Report

Comments and Suggestions for Authors

The reviewed paper "Seismic Monitoring of a Deep Geothermal Field in Munich

(Germany) using Borehole Distributed Acoustic Sensing" concerns with a relatively new method of measuring seismic signals by the distributed sensors installed in boreholes. The paper is interesting for a potential reader of the journal, and can be recommended for publication, after a minor improvement.

1. The authors assert that the DAS eliminate seismic noise caused by the so called anthropogenic seismicity. It is of course true, because deeply installed DAS do not record signals from short-period SAW, which rapidly attenuate with depth, while most of anthropogenic noise propagate in the form of surface acoustic waves associated with Rayleigh (Rayleigh-Lamb) and more rare Love waves. But, there are other methods for filtering high-frequency noise, especially produced by anthropogenic sources. In this respect, the short-period anthropogenic noise can be separated and filtered out by the use of Rayleigh formula for the ratio between vertical and horizontal displacement components; this ratio depends on Poisson's ratio, and does not depend upon frequency. This result was later on extrapolated to Rayleigh-Lamb propagating in stratified and functionally-graded media; e.g. DOI: 10.1134/S1061830917040039, enabling to eliminate both vertical and horizontal noise components.

Author Response

The reviewed paper "Seismic Monitoring of a Deep Geothermal Field in Munich (Germany) using Borehole Distributed Acoustic Sensing" concerns with a relatively new method of measuring seismic signals by the distributed sensors installed in boreholes. The paper is interesting for a potential reader of the journal, and can be recommended for publication, after a minor improvement. The authors assert that the DAS eliminate seismic noise caused by the so called anthropogenic seismicity. It is of course true, because deeply installed DAS do not record signals from short-period SAW, which rapidly attenuate with depth, while most of anthropogenic noise propagate in the form of surface acoustic waves associated with Rayleigh (Rayleigh-Lamb) and more rare Love waves. But, there are other methods for filtering high-frequency noise, especially produced by anthropogenic sources. In this respect, the short-period anthropogenic noise can be separated and filtered out by the use of Rayleigh formula for the ratio between vertical and horizontal displacement components; this ratio depends on Poisson's ratio, and does not depend upon frequency. This result was later on extrapolated to Rayleigh-Lamb propagating in stratified and functionally-graded media; e.g. DOI: 10.1134/S1061830917040039, enabling to eliminate both vertical and horizontal noise components

>> Thank you for your revision and for your feedback. We have considered your comment and the inclusion of the publication you suggest in the bibliography. However, we feel that the publication is of little relevance given our case study. Indeed, DAS is a technology known for its unidirectional sensitivity and provides measurements of strain-rate (or strain) along a single component, specifically the axial length of the fiber.

In our context, only the vertical component is recorded due to the vertical orientation of the TH3 well. As a result, de-noising methods that rely on the combined use of vertical and horizontal displacement components are not applicable.

Therefore, we preferred not to refer to this method in the manuscript.

Reviewer 4 Report

Comments and Suggestions for Authors

Thank you for your hard work.

The paper is very well written and contains a description of the long-term observation. The obtained results are very important in the presented field of science.

There are only a few editorial problems:

1. The numbering of formulas is too close to them. Those numbers have to be on the right side of the page when the formulas need to be centered.

2. In the bibliography, the following positions: 1, 30, 40, and 57 are not completed. It is not clear if the publications are regular paper, books, or other kinds. It has to be corrected.

Author Response

Thank you for your hard work. The paper is very well written and contains a description of the long-term observation. The obtained results are very important in the presented field of science. There are only a few editorial problems:

Thank you for your review and for your positive feedback. We addressed the remaining issues and respond to your comments in the following (in red).

  1. The numbering of formulas is too close to them. Those numbers have to be on the right side of the page when the formulas need to be centered.

>> The style for equations was modified in Eq. 1 to Eq. 5 (see Section 3.2.2).

  1. In the bibliography, the following positions: 1, 30, 40, and 57 are not completed. It is not clear if the publications are regular paper, books, or other kinds. It has to be corrected

>> We completed the positions 1, 30, 40 (now, 41). The reference associated with [57] was removed because the paragraph was revised according to other reviewer comments.

Reviewer 5 Report

Comments and Suggestions for Authors

The paper utilizes Distributed Acoustic Sensing (DAS) technology for seismic monitoring of deep geothermal fields, which is a highly innovative research direction. The study fully exploits the potential of DAS technology in monitoring microseismic activities in urban geothermal fields. By comparing the data with that from traditional seismometers, it demonstrates the reliability and efficiency of DAS. However, there are still numerous issues with the paper that need to be addressed:

1. The paper describes the P-waves and S-waves received by DAS, but the actual signals are not P-waves or S-waves. It needs to explain how to derive P-waves and S-waves from DAS signals.

2. Although the methods section is generally strong, the description of the process of converting DAS strain data into acceleration is not detailed enough. The paper should further explore the impact of this conversion on the accuracy of seismic event characterization and provide more technical details or verification processes to enhance the credibility of the results.

3. The paper lacks detailed descriptions of the filtering algorithms used. It should include specific formulas to clearly illustrate how the filtering is implemented. This would help in understanding the modifications made to the raw data and how these affect the final interpretation of seismic signals.

4. The statistical analysis in the paper could be enhanced by adding quantification of data uncertainty. Providing statistical metrics or error bounds for seismic event detection and description would allow for a deeper understanding of the monitoring system's reliability. This approach would not only strengthen the validity of the results but also help in assessing the precision and robustness of the DAS technology in seismic applications.

5. The paper mentions the importance of an accurate velocity model but does not discuss in detail the sources of the current velocity models and their potential limitations. Expanding the discussion on the methods for establishing velocity models, as well as any discrepancies observed during application, would significantly enhance the value of the paper. This deeper analysis could lead to a better understanding of the impact of velocity model accuracy on seismic monitoring results and could suggest areas for further research or improvement.

6. The discussion section could be further expanded to include the environmental and operational impacts of implementing the DAS system in urban geothermal fields. While the technical capabilities have been detailed, including discussions on its sustainability, cost-effectiveness, and potential impacts on urban infrastructure would make the paper more comprehensive. This broader perspective would help stakeholders better understand the practical implications and feasibility of deploying DAS technology in urban settings.

Author Response

The paper utilizes Distributed Acoustic Sensing (DAS) technology for seismic monitoring of deep geothermal fields, which is a highly innovative research direction. The study fully exploits the potential of DAS technology in monitoring microseismic activities in urban geothermal fields. By comparing the data with that from traditional seismometers, it demonstrates the reliability and efficiency of DAS. However, there are still numerous issues with the paper that need to be addressed:

Thank you for your review and your feedback, which contribute to improve the paper. A point-by-point answer to each of your comments is proposed in the following (in red).

  1. The paper describes the P-waves and S-waves received by DAS, but the actual signals are not P-waves or S-waves. It needs to explain how to derive P-waves and S-waves from DAS signals.

>> Sure, DAS does not record directly P- (compressional) and S- (shear) waves, as it senses the relative deformation of tiny portion of optical fibers.

As pointed out in the before-last paragraph of the introduction, recent studies suggest the validity of the assumption of no phase or minimal phase distortion in DAS data, compared with conventional seismometer measurements. Therefore, the onset-times of the signature of the P- and S-waves on the DAS data (i.e. on the strain-rate “trace”) correspond to the onset-times of the P- and S-seismic waves. For sake of simplicity, as you noticed, we made a misused of language and sometimes talked about seismic waves seen on the DAS. To avoid any confusion, we decided to emphasize that DAS records the signature of the P- and S-waves (e.g. caption of Fig. 3).

  1. Although the methods section is generally strong, the description of the process of converting DAS strain data into acceleration is not detailed enough. The paper should further explore the impact of this conversion on the accuracy of seismic event characterization and provide more technical details or verification processes to enhance the credibility of the results.

>> In the manuscript, we refer to the original paper that introduced and detailed the procedure (Lior et al, 2019), and we mention that the procedure was validated in this reference paper too (subsection 3.2.2. after equation 1). The subsection 5.2 about the assessment of the DAS-based acceleration covers aspects of the reliability of the conversion procedure with a real event.

  1. The paper lacks detailed descriptions of the filtering algorithms used. It should include specific formulas to clearly illustrate how the filtering is implemented. This would help in understanding the modifications made to the raw data and how these affect the final interpretation of seismic signals.

>> We added details about the filtering procedure in the first paragraph of subsection 3.1. In particular, we describe the Butterworth bandpass filter. We also give further details about the f-k filter designed for the study.

  1. The statistical analysis in the paper could be enhanced by adding quantification of data uncertainty. Providing statistical metrics or error bounds for seismic event detection and description would allow for a deeper understanding of the monitoring system's reliability. This approach would not only strengthen the validity of the results but also help in assessing the precision and robustness of the DAS technology in seismic applications.

>> It is not possible to compute robust statistics about the detection capability of the DAS system, considering the limited seismicity observed in the study zone and /or the duration of the study.The question of the detection capabilities was already addressed in Azzola et al. (2023) (position [17] in the manuscript).

Regarding the processing of the seismic events, the difference between DAS-based data and seismometer-based data is the conversion from strain-rate (of DAS data) to acceleration. Addressing the question of the robustness of DAS in seismic applications therefore requires assessing the reliability of this additional conversion. In subsection 5.2, we address this point by comparing waveforms with the closest seismometer (SYBAD). Unfortunately, this seismometer is not collocated to the FOC (, which would require cementation of a seismometer behind the casing). Hence, computing error bounds on the DAS-based acceleration values is not possible.

Nevertheless, we analyzed and quantified the contribution of the DAS and compared results with and without it (section 5 “Discussion”). For example, we showed that the information carried by the DAS is valuable and reliable, and may be more reliable than standard surface 3C-seismometers for the seismic source parameters quantification (e.g. Figure 8).

Finally, an evaluation and comparison of DAS instrument responses across various settings is presented by Paitz et al. (2020), for instance. This reference, which is cited in the introduction and in section 5.2, underlines the reliability of DAS technology in seismic applications requiring the complete waveform, including their phase and amplitude information.

Paitz, P., Edme, P., Gräff, D., Walter, F., Doetsch, J., Chalari, A., Schmelzbach, C., et al., 2020. Empirical Investigations of the Instrument Response for Distributed Acoustic Sensing (DAS) across 17 Octaves. Bulletin of the Seismological Society of America, 111, 1–10. doi:10.1785/0120200185

  1. The paper mentions the importance of an accurate velocity model but does not discuss in detail the sources of the current velocity models and their potential limitations. Expanding the discussion on the methods for establishing velocity models, as well as any discrepancies observed during application, would significantly enhance the value of the paper. This deeper analysis could lead to a better understanding of the impact of velocity model accuracy on seismic monitoring results and could suggest areas for further research or improvement.

>> The procedure for the construction of the velocity model is detailed in Appendix B. For clarity purposes, we better referenced this appendix in the main part of the manuscript.

We developed the last paragraph of subsection 5.1 to broaden discussion of the methods used to develop and calibrate velocity models, in order to build confidence in location results.

The influence of the velocity model on seismic event location is a significant seismological topic, albeit not the focus of this manuscript. The analysis, which is already addressed by section 5.1, was therefore not extended. It is also important to note that the detection procedure implemented does not depend on any velocity model (see the statement added in subsection 3.1).

  1. The discussion section could be further expanded to include the environmental and operational impacts of implementing the DAS system in urban geothermal fields. While the technical capabilities have been detailed, including discussions on its sustainability, cost-effectiveness, and potential impacts on urban infrastructure would make the paper more comprehensive. This broader perspective would help stakeholders better understand the practical implications and feasibility of deploying DAS technology in urban settings

>> Although your proposal of including environmental, operational, sustainability and cost-effectiveness of DAS for seismic monitoring is very interesting, it would necessitate a thorough environmental, risk and economical study of the numerous factors involved. It would require gathering data in various geothermal contexts, which could constitute a research topic on its own. Hence, this is beyond the scope of our manuscript.

Round 2

Reviewer 5 Report

Comments and Suggestions for Authors

I have received all my feedback and have no further comments.